# Using Reinforcement Learning to Train Large Language Models to Explain Human Decisions

**Jian-Qiao Zhu**[*]
Princeton University and The University of Hong Kong
zhujq@hku.hk

**Hanbo Xie**[*]
Georgia Institute of Technology

**Dilip Arumugam**
Princeton University

**Robert C. Wilson**[†]
Georgia Institute of Technology

**Thomas L. Griffiths**[†]
Princeton University

## Abstract

A central goal of cognitive modeling is to develop models that not only predict human behavior but also provide insight into the underlying cognitive mechanisms. While neural network models trained on large-scale behavioral data often achieve strong predictive performance, they typically fall short in offering interpretable explanations of the cognitive processes they capture. In this work, we explore the potential of pretrained large language models (LLMs) to serve as dual-purpose cognitive models — capable of both accurate prediction and interpretable explanation in natural language. Specifically, we employ reinforcement learning with outcome-based rewards to guide LLMs toward generating explicit reasoning traces that explain human risky choices. Our findings demonstrate that this approach produces high-quality explanations at scale alongside strong quantitative predictions of human decisions.

## 1 Introduction

Computational models of cognition have driven immense progress in understanding the mental processes underlying learning, thinking, problem solving, and decision making (Farrell & Lewandowsky, 2018; Busemeyer & Diederich, 2010; Griffiths et al., 2024a; Ma et al., 2023; Rumelhart et al., 1986). Recent advances — particularly those leveraging neural networks to predict human behavior — have introduced increasingly sophisticated model architectures and training methods (Peterson et al., 2021; Zhu et al., 2025a; Huang, 2023; Almaatouq et al., 2024). While these models have demonstrated improved predictive accuracy, they often lack interpretability, offering limited insight into the underlying cognitive mechanisms. A recent example is the Centaur model, which applies customized supervised fine-tuning (SFT) to pretrained LLMs and achieves impressive performance in predicting human behavior, surpassing domain-specific cognitive models from the literature (Binz et al., 2025). However, despite their predictive successes, such models offer limited explanatory power; a deeper theoretical understanding of human cognition requires more than a simple match in behavior (Frank & Goodman, 2025).

To start addressing this dilemma, we propose that LLMs capable of generating reasoning or thinking tokens offer a promising opportunity for developing cognitive models that not only *predict* but also *explain* human behavior. The key idea is to treat the chain-of-thought (CoT) generated by these models prior to their final responses as a verbalized account of underlying cognitive mechanisms, expressed in natural language. In other words, while LLMs learn to generate CoT reasoning that improves prediction of human behavior, cognitive scientists can examine these CoTs to assess whether they provide meaningful and interpretable explanations of the observed data.

In this work, we put these ideas to the test in the domain of human risky choice; that is, how people choose between risky and/or safe options. To encourage LLMs to generate both CoTs and predictions about human risky choices, we post-train a backbone LLM using reinforcement learning

---

[*]Equal contribution
[†]Equal advising

(RL) with outcome-based rewards (Lambert, 2025; Shao et al., 2024; Yu et al., 2025; Liu et al., 2025), where human choice proportions served as the ground truth for evaluation. In other words, the behavioral predictions are directly linked to the reward function of RL, incentivizing the LLM to produce useful reasoning chains. For comparison, we also evaluated various SFT post-training methods, including Centaur-style SFT (Binz et al., 2025), which has been shown to produce high-performing cognitive models based on LLMs.

Our findings show that RL post-training can elicit sensible CoT reasoning traces from LLMs while achieving predictive accuracy comparable to that of SFT-based methods. Moreover, the generated CoTs are responsive to the structure of the training data: when human behavioral data are replaced with synthetic data generated by an expected-value maximization model, the CoTs adapt accordingly to reflect the structure of the synthetic dataset. We also find that the quality of the CoTs depends on the strength of the backbone LLM and thus using a weaker model results in noticeable degradation of reasoning quality.

## 2 RELATED WORK

**Leveraging LLMs to model human cognition.** In recent years, there has been widespread enthusiasm about the potential of LLMs to advance cognitive modeling and provide new theoretical understanding about the mind. A growing body of work has sparked scholarly debate around the use of LLMs for modeling human cognitive processes (Frank & Goodman, 2025; Griffiths et al., 2024b; Binz & Schulz, 2023). The main advantage of using LLMs to predict human behavior is that LLMs can process similar stimuli to people; in other words, LLMs can process a broader range of stimuli (often described in natural language) than previous neural network models which typically operate in a more abstract representation of cognitive tasks (Frank & Goodman, 2025; Peterson et al., 2021; Huang, 2023; Zhu et al., 2025a).

Consider the risky-choice problem illustrated in Figure 1, where human participants were asked to choose between Option A, which offers $27 for sure, and Option B, which offers $25 with a 90% probability and $92 with a 10% probability. Traditional cognitive models and neural-network-based models typically operate on structured quantitative task features such as $\{1, \$27\}$ (i.e., probability and value) for option A and $\{0.9, \$25; 0.1, \$92\}$ for option B (Peterson et al., 2021). These models define functions that map from such numerical inputs to human risky choices. In contrast, LLM-based cognitive models operate over a different representation of the task: they take natural language descriptions as input and predict human responses directly (Zhu et al., 2025b; Binz et al., 2025; Binz & Schulz, 2023). Recent work has shown that SFT of LLMs can improve predictive accuracy on human behavioral data (Binz et al., 2025). Our work goes beyond improving LLMs' ability to predict human behavior; we aim to elicit verbal theories of human behavior from LLMs using RL.

**Automated discovery of cognitive models using LLMs.** Another closely-related line of research involves using LLMs to automatically search over the space of cognitive models, enabling the automated discovery of interpretable theories (Castro et al., 2025; Rmus et al., 2025; Musslick et al., 2024; Wong et al., 2023). In this approach, LLMs are prompted to generate symbolic programs (e.g., Python code), which are then executed and fitted to human or animal data. Because the discovered models are expressed as code, they are inherently interpretable — helping to address the aforementioned dilemma in cognitive modeling. Leveraging the coding capabilities of LLMs has shown promise in identifying heuristic decision-making models (Rmus et al., 2025) and strategies in multi-armed bandit tasks (Castro et al., 2025). However, these approaches typically do not involve further fine-tuning of the LLM during the search process, instead relying heavily on the model's in-context learning (Brown et al., 2020) ability for effective model discovery.

## 3 METHOD

To evaluate whether post-training LLMs can produce useful cognitive models, we compare three distinct post-training strategies for LLMs: (i) SFT, (ii) a variant of SFT specifically designed for adapting LLMs to cognitive tasks, as used in the Centaur model (Binz et al., 2025), and (iii) RL based on Group Relative Policy Optimization (GRPO) (Shao et al., 2024; Liu et al., 2025). Each method was applied to fine-tune identical low-rank adaptation (LoRA) modules on the largest available human risky-choice dataset, `choices13k`, originally collected by Peterson et al. (2021).

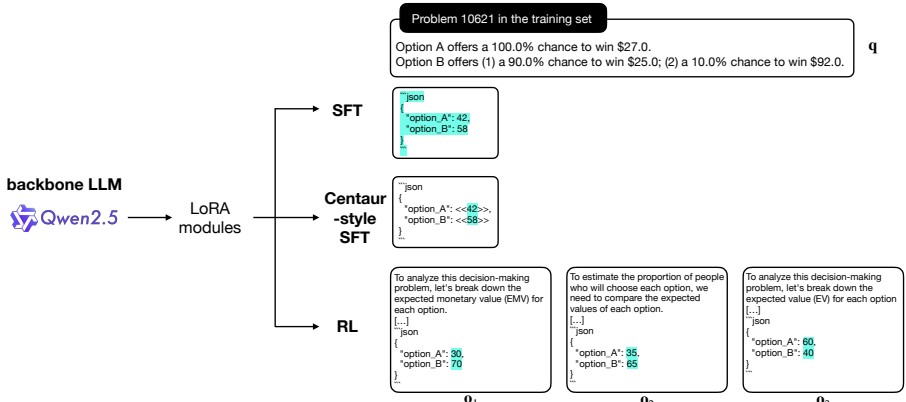

Figure 1: Overview of three post-training strategies for building cognitive models of human risky choice using Qwen-2.5-7B-Instruct. The backbone LLM was first adapted using low-rank adaptation (LoRA) (Hu et al., 2022), followed by post-training via three strategies: supervised fine-tuning (SFT), Centaur-style SFT (Binz et al., 2025), and reinforcement learning from outcome-based rewards (Liu et al., 2025; Shao et al., 2024). In the illustrated example, the LLM is prompted to predict human choice behavior. SFT and Centaur-style models are trained directly on human data represented in JSON format. In contrast, the RL model generates candidate completions that include CoT reasoning and final predictions in JSON format, with each completion evaluated based on its predictions. Tokens or predictions relevant to each training method are highlighted in light blue.

The backbone LLM used throughout is Qwen-2.5-7B-Instruct (Yang et al., 2024). All models were fine-tuned using the same LoRA configuration (Hu et al., 2022), with both the rank and alpha parameters set to 32. LoRA modules were applied to all linear layers of the backbone model and included a dropout probability of 0.05. This setup results in approximately 80.74M trainable parameters, comprising only 1.05% of the total parameters in the 7B model.

To ensure consistency and control across training and evaluation, we randomly partitioned the `choices13k` dataset into training and test sets using a 90/10 split, resulting in 13,102 choice problems for training and 1,462 for testing. The partition was performed at the level of unique problem identifiers, such that repeated measures of the same problem, if present, were assigned entirely to either the training or the test set. This partition was held fixed for all post-training, including SFT, Centaur-style SFT, and RL, as well as evaluation procedures.

**SFT and Centaur-style SFT training details.** We begin with the standard SFT approach to adapt a pretrained LLM to the target domain of human risky choice behavior (see Figure 1 SFT). To enhance the Qwen model's ability to predict human decisions, we converted the original `choices13k` dataset into a text format. Specifically, we represented human choice data in JSON, rounding the observed choice percentages for each option to the nearest integer between 0 and 100. For example, if empirical data show that 71.11% of participants selected Option B, this was reformatted as JSON: {"option_A": 29, "option_B": 71}. See Appendix A.1 for detailed prompts.

We fine-tuned the Qwen model using SFT for a total of 6 epochs on the training set, with a fixed learning rate of $10^{-5}$, gradient accumulation steps of 8, and the AdamW optimizer (Loshchilov & Hutter, 2017). In our setup, SFT corresponds to the standard autoregressive next-token prediction paradigm.

In parallel, we implemented a variant used to train the Centaur model, which selectively masks out all tokens except those corresponding to human data (Binz et al., 2025). This method has been shown to effectively adapt pretrained LLMs for explaining a wide range of human cognitive tasks (Binz et al., 2025). Unlike standard SFT, Centaur-style SFT places human data (i.e., choice proportions) within special brackets ("<<" and ">>"), and only tokens enclosed within these brackets contribute to the loss during training (see Figure 1). In our implementation, Centaur-style SFT was applied at the problem level rather than the individual participant level, such that the model predicts aggregated human responses for each risky-choice problem.

**RL training details.** We also fine-tuned the Qwen model using the GRPO algorithm (Shao et al., 2024). At each training step, the model generates 12 candidate completions (i.e., $\mathbf{o}_1, \mathbf{o}_2, ..., \mathbf{o}_G$) for the same risky-choice problem (i.e., $\mathbf{q}$), with each completion capped at a maximum length of 1,024 tokens. The GRPO algorithm evaluates each candidate based on the model-predicted choice probabilities for the two options (see Figure 1 RL). Specifically, we implemented an outcome-based reward function that uses human data as ground truth. The reward is defined as one minus the absolute difference between the model's predicted probability for option B and the empirical proportion of human participants who chose that option:

$$R(\mathbf{q}, \mathbf{o}) = \begin{cases} 1 - |\mathbf{o}^B - \mathbf{p}^B| & \text{if } \mathbf{o} \text{ is coherent} \\ 0 & \text{otherwise} \end{cases} \tag{1}$$

Here, $\mathbf{q}$ denotes the risky-choice problem, $\mathbf{o}$ is the model response, $\mathbf{o}^B$ is the model-predicted probability for option B, and $\mathbf{p}^B$ is the proportion of human participants who selected option B. To be eligible for a reward, model-predicted probabilities for both option A and B must satisfy coherence constraints: $0 \leq \mathbf{o}^A \leq 1$, $0 \leq \mathbf{o}^B \leq 1$, and $\mathbf{o}^A + \mathbf{o}^B = 1$. The maximum achievable outcome reward is 1, which occurs when $\mathbf{o}^B = \mathbf{p}^B$.

In addition, we incorporated a format reward to encourage structurally well-formed completions. This reward depends on the number and position of JSON-formatted outputs in the model response. If the model generates exactly one JSON output, the format reward increases by 0.25. Furthermore, if the behavioral prediction appears after reasoning tokens, the format reward increases by an additional 0.25. The maximum possible format reward is therefore 0.5.

The RL training was conducted for a total of 3 epochs, with a learning rate of $3 \times 10^{-6}$ and a cosine learning rate scheduler. Moreover, we omitted the reward normalization by standard deviation as suggested by the GRPO Done Right algorithm (Liu et al., 2025). As a result, the advantage function was defined as the reward of each candidate completion minus the average reward within the group.

$$A_i = R(\mathbf{q}, \mathbf{o}_i) - \text{mean}(\{R(\mathbf{q}, \mathbf{o}_1), ..., R(\mathbf{q}, \mathbf{o}_G)\}) \tag{2}$$

where $G = 12$ is the group size. We revisit the implications of this design choice, along with other unsuccessful attempts, in the Discussion section and Appendix F.

With the advantage function defined, we train the backbone LLM using the GRPO Done Right algorithm (Liu et al., 2025) with the following objective function:

$$\mathcal{L}(\pi_\theta) = \mathbb{E}_{\mathbf{q} \sim Q, \{\mathbf{o}_i\}_{i=1}^G \sim \pi_{\theta_{\text{old}}}(\cdot|\mathbf{q})}$$

$$\frac{1}{G} \sum_{i=1}^G \sum_{t=1}^{|\mathbf{o}_i|} \left\{ \min \left[ \frac{\pi_\theta(o_{i,t}|\mathbf{q}, \mathbf{o}_{i,<t})}{\pi_{\theta_{\text{old}}}(o_{i,t}|\mathbf{q}, \mathbf{o}_{i,<t})} A_i, \text{clip}\left( \frac{\pi_\theta(o_{i,t}|\mathbf{q}, \mathbf{o}_{i,<t})}{\pi_{\theta_{\text{old}}}(o_{i,t}|\mathbf{q}, \mathbf{o}_{i,<t})}, 1 - \epsilon_{\text{low}}, 1 + \epsilon_{\text{high}} \right) A_i \right] \right\} \tag{3}$$

where $\mathbf{q}$ denotes a risky-choice problem sampled from the training set $Q$, and $|\mathbf{o}_i|$ is the number of tokens in the i-th model completion $\mathbf{o}_i$, generated by the LLM under the old policy $\pi_{\theta_{\text{old}}}$. Moreover, $t$ denotes the position of a token within the sequence of a model-generated completion and $\theta$ represents the parameters of LLM. The inner summation iterates over all tokens in the sampled completion. This objective follows the clipped surrogate loss formulation from the proximal policy optimization (PPO) algorithm (Schulman et al., 2017), modified to operate at the token level within each sampled trajectory. The clipped values lie within the range of [0.8, 1.28], where $\epsilon_{\text{low}} = 0.2$ and $\epsilon_{\text{high}} = 0.28$, as set in our experiment. This asymmetric clipping follows the recommendation in Yu et al. (2025), which suggests that slightly increasing $\epsilon_{\text{high}}$ can enhance exploration in RL. GRPO also incorporates a KL divergence penalty term in its objective function to prevent the updated policy from deviating too far from the backbone LLM. This penalty takes the form $\beta D_{KL}(\pi_\theta \parallel \pi_{\text{reference}})$, where $\beta$ is set to $10^{-4}$.

**Key distinctions between RL and Centaur-style SFT.** While both Centaur-style SFT and RL focus exclusively on tokens relevant to choice probabilities (see Figure 1, highlighted text), there are important distinctions between these two post-training methods. Centaur-style SFT operates within the standard next-token prediction framework and thus relies on the tokenized representation of numerical outputs. In contrast, RL assigns outcome rewards based on the predicted numerical values themselves, rather than their tokenized forms. These rewards are then used to weight policy updates during training. This reward-weighted policy optimization has been argued to support improved generalization in downstream tasks (Chu et al., 2025; Wang et al., 2025).

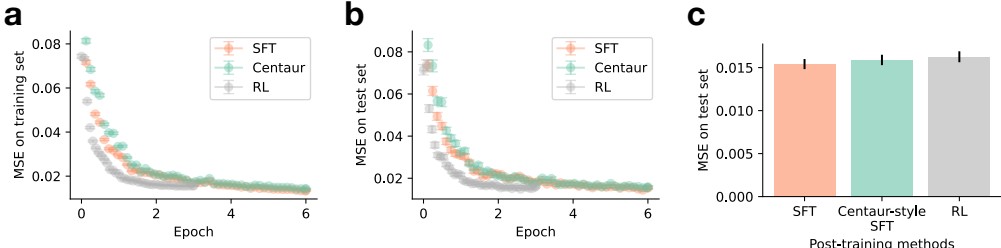

Figure 2: Learning curves on the **(a)** training and **(b)** test sets. Backbone LLM is Qwen-2.5-7B-Instruct. The horizontal axes indicate training epochs, while the vertical axes represent mean squared error (MSE) evaluated on the corresponding dataset. The three post-training strategies compared are supervised fine-tuning (SFT, red), Centaur-style SFT (green), and reinforcement learning (RL, grey). The lowest MSEs on the training set occur at epochs 5.98 (SFT), 5.74 (Centaur-style SFT), and 2.75 (RL), while the lowest MSEs on the test set occur at epochs 5.86 (SFT), 5.86 (Centaur-style SFT), and 2.60 (RL). **(c)** MSE on the test set at the final checkpoint of each post-training method. Error bars represent $\pm 1$ standard error across risky-choice problems.

## 4 RESULTS

Having introduced the three post-training strategies, we now evaluate the effectiveness of the fine-tuned models from each method as cognitive models of human risky choice. To ensure comparability, all fine-tuned models were evaluated using identical sampling parameters, implemented with vLLM (Kwon et al., 2023): temperature = 0.7, top-p = 0.95, and top-k = 50. The maximum number of generated tokens was adjusted according to model type. Since the RL models were trained to generate intermediate reasoning before producing a final choice prediction, they were allowed up to 1,024 tokens to accommodate more elaborate completions. In contrast, the SFT and Centaur-style models were restricted to 30 tokens, as they were trained to produce choice predictions directly without intermediate reasoning steps. We evaluated model inferences at all checkpoints across the three post-training methods, for both the training and test set problems.

**Learning curves.** All three post-training methods led to gradual improvements in model predictions, as indicated by decreasing MSE on both the training and test sets (see Figure 2). However, the error trajectories differ notably across methods. RL achieves a faster reduction in prediction error compared to SFT and Centaur-style SFT when measured by the number of training examples processed. It is important to note, however, that RL generates significantly more completions per training example and incurs substantially higher computational costs, whereas SFT and Centaur-style SFT are trained directly on the human data without generating additional completions.

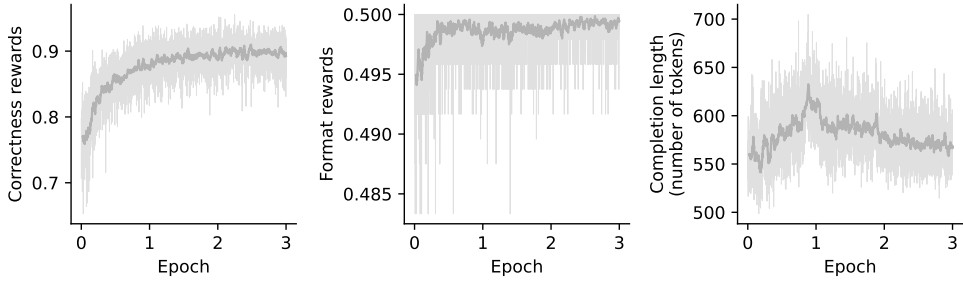

Figure 3: Learning curves for the RL model. Backbone LLM is Qwen-2.5-7B-Instruct. (*Left*) Correctness reward, defined as one minus the absolute difference between model predictions and human choice proportions. (*Middle*) Format reward, based on the structure and position of the model's JSON output. (*Right*) Completion length, measured by the number of generated tokens.

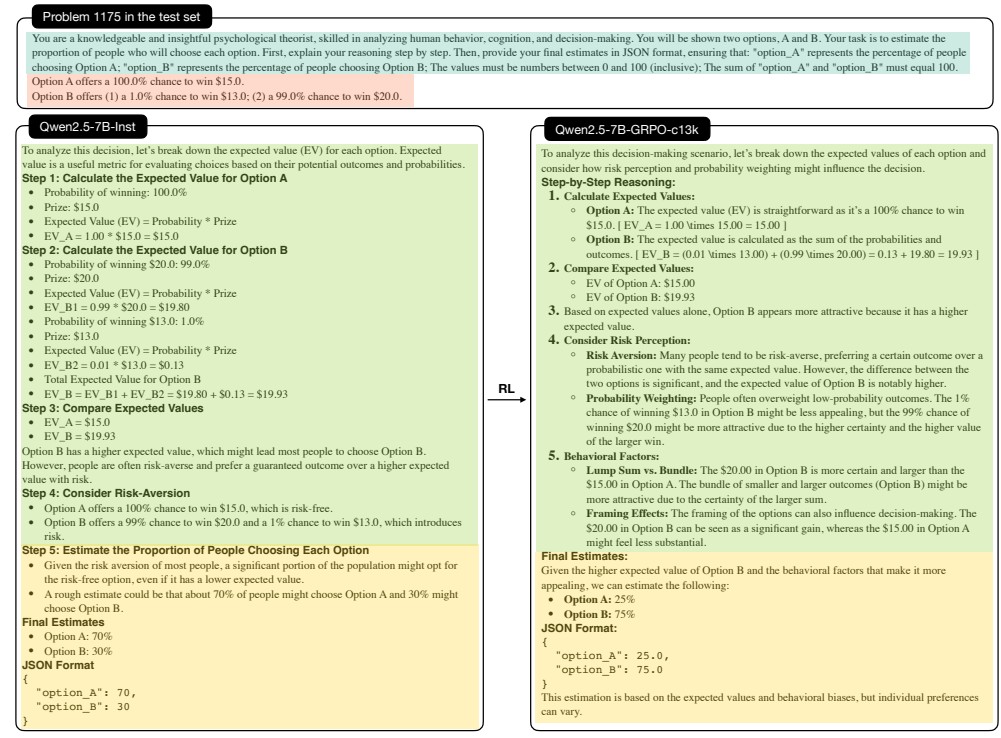

Figure 4: Comparison of CoT reasoning and model predictions before (left panel) and after (right panel) RL training. Human data indicate that approximately 71% of participants selected Option B in this problem. Model completions are reformatted using Markdown for enhanced readability.

Ultimately, all three post-training methods yield fine-tuned models with comparable performance in predicting human choices. At the final checkpoint, the MSE on test set is $M = .0144$ ($SE = .0006$) for SFT, $M = .0155$ ($SE = .0006$) for Centaur-style SFT, $M = .0148$ ($SE = .0006$) for RL. Statistical tests reveal no significant differences among the models: SFT vs. Centaur-style SFT ($t(2923) = -1.31, p = 0.19$), SFT vs. RL ($t(2923) = -0.58, p = 0.56$), and RL vs. Centaur-style SFT ($t(2923) = 0.78, p = 0.43$). For reference, the best-performing neural network model reported in (Peterson et al., 2021), the Mixture of Theories model, achieved an MSE of .0113, while the neurally augmented prospect theory model achieved an MSE of .0204 (both models use problem features as input rather than text).

Moreover, we observe stable improvements in the RL learning curves for both the correctness reward and the format reward (see Figure 3). The format reward is learned rapidly, with the ceiling value of 0.5 reached within the first few training steps. The completion length of the RL model initially increases during the first training epoch and subsequently stabilizes between 500 and 650 tokens.

**Analyzing Chains-of-Thought.** Unlike the SFT and Centaur-style models, RL models are capable of explicitly generating reasoning tokens prior to making final predictions. We analyze these reasoning chains with three key objectives: (i) to characterize the nature of the reasoning processes across different risky-choice problems and training stages; (ii) to assess whether the reasoning chains exhibit causal influence on the model's predictions; and (iii) to examine the implications of these reasoning traces for developing cognitive models of human risky choice.

As illustrated in Figure 4, RL training alters both the CoT reasoning and final predictions of the LLM for the same problem in the held-out test set. We observe that certain thoughts are preserved: for example, the calculation and comparison of expected values for both options, as well as considerations of human risk preferences. In addition, RL appears to amplify certain types of thoughts in the CoT, such as references to psychological factors and cognitive biases.

We next aim to characterize the CoT reasoning learned by the RL model in order to identify effective cognitive processes that contribute to accurately predicting human risky choices. Note that the CoT

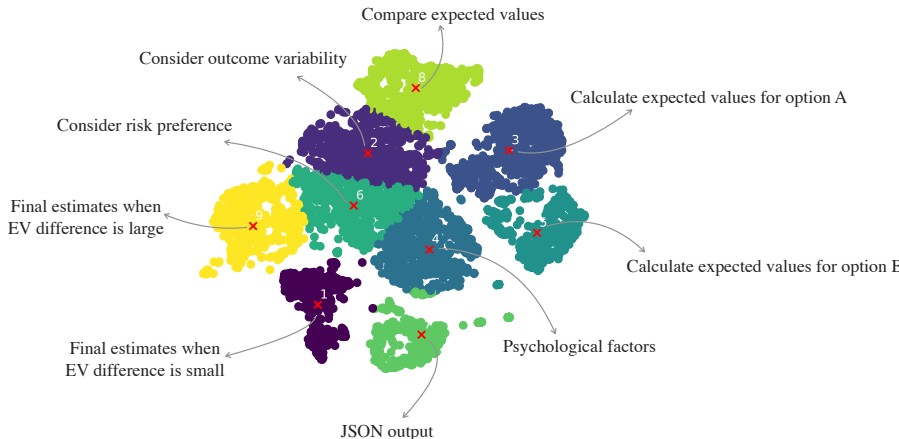

Figure 5: Visualization of CoT reasoning generated by RL models. Each reasoning segment ("thought") is embedded using the `all-MiniLM-L6-v2` model from SBERT (Reimers & Gurevych, 2019), followed by dimensionality reduction to two dimensions using t-SNE (Van der Maaten & Hinton, 2008). In the resulting 2D space, we identified nine clusters using the k-means algorithm. Each cluster is labeled with a summary of its centroid thought to provide an interpretable overview of common reasoning patterns.

generated by the RL model is typically itemized, often presented as a sequence of clearly delineated steps or sections. Leveraging this structure, we segmented each CoT into smaller units — referred to as thoughts — using a regular expression-based parser. For example, the CoT shown in Figure 5 (right) is divided into five distinct thoughts, corresponding to its five itemized sections.

Each segmented thought was embedded using the `all-MiniLM-L6-v2` model from SBERT (Reimers & Gurevych, 2019), and subsequently projected into two dimensions using t-SNE for visualization (Van der Maaten & Hinton, 2008). In this 2D embedding space, we identified nine clusters using the k-means algorithm. To enhance interpretability, each cluster was labeled with a summary of its centroid thought, providing an overview of common reasoning patterns.

As shown in Figure 5, the CoT generated by the RL model can be broadly categorized into five types of reasoning strategies that contribute to improved prediction of human risky choices: (i) computing the expected values of both options accurately, (ii) conducting a coarse comparison based on these expected values, (iii) considering psychological influences such as behavioral biases, (iv) incorporating human risk preferences and the variability of outcomes, and (v) producing a final prediction based on some or all of the above factors, conditional on the magnitude of the expected value difference.

Moreover, we conducted several supplementary analyses of CoT reasoning, including small-scale human evaluations (Appendix C.1), large-scale LLM-as-a-judge assessments (Appendix C.2), CoT swapping between the backbone and RL models (Appendix C.3), and tests of the effectiveness of SFT-generated and Centaur-generated CoTs (Appendix C.4). Taken together, these analyses suggest that RL post-training improves the quality of CoT reasoning, which in turn contributes to enhanced prediction accuracy of human behavior.

**Cognitive mechanisms identified in CoT.** We examined the cognitive mechanisms reflected in the RL model's CoT reasoning. In our context, cognitive mechanisms are specifically defined as verbal theories that explain how people reach a risky decision when presented with a particular set of options. To assist with this analysis, we used OpenAI's GPT-4.1 (`https://openai.com/index/gpt-4-1/`) to summarize the CoT outputs generated by the RL model (see Appendix B for detailed prompts). The evolution of the top eight cognitive mechanisms over training epochs is shown in Figure 6a. Among these, expected value computation and risk aversion emerged as the two most frequently used mechanisms, each accounting for approximately 29% to 36% of thoughts across risky-choice problems.

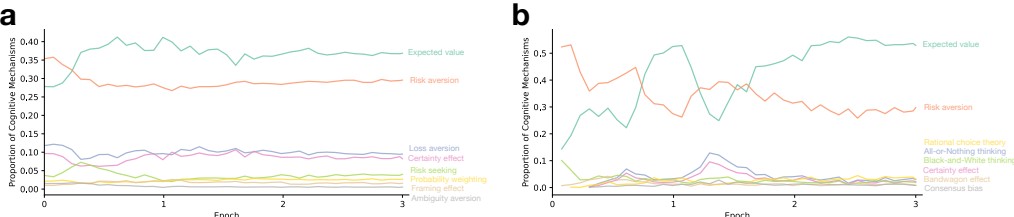

Figure 6: Proportions of cognitive mechanisms identified in the RL models' thoughts across training epochs. Backbone LLM is Qwen-2.5-7B-Instruct. The top eight most frequently used mechanisms are displayed. **(a)** Training data are human risky choices. **(b)** Training data are synthetic choices generated by an expected-value maximization model.

We observed notable representations of loss aversion — the tendency for individuals to prefer avoiding losses over acquiring equivalent gains (Tversky & Kahneman, 1992) — and the certainty effect, where individuals disproportionately favor certain outcomes over those that are merely probable (Cohen & Jaffray, 1988). Finally, the RL model occasionally considers the possibility that some individuals may exhibit risk-seeking behavior or be influenced by other cognitive biases, such as probability weighting (Prelec, 1998; Kahneman & Tversky, 2013), framing effects (Tversky & Kahneman, 1981), and ambiguity aversion (Fox & Tversky, 1995). The full list of cognitive mechanisms identified by the RL model is provided in Table 1 and Appendix B.

These results also provide an interesting perspective for theorists working on models of human risky choice, a domain where substantial research effort has been devoted to identifying systematic deviations from rationality and developing heuristics and biases to account for human irrational behaviors (Tversky & Kahneman, 1992; Kahneman & Tversky, 2013; Gigerenzer & Gaissmaier, 2011; Gilovich et al., 2002). In contrast, the RL model highlights calculating expected values and risk aversion as the dominant forces driving the explanation and prediction of human risky choices. This finding suggests that greater attention should be paid to rational characterizations of human behavior and that there may be value in developing new theories grounded in rational principles. Recent work in this area suggests that human decisions may be explained in terms of the rational use of cognitive resources (Gershman et al., 2015; Lieder & Griffiths, 2020; Zhu & Griffiths, 2025).

## 5 CONTROL EXPERIMENTS

**Data control.** To further validate the RL method, we conducted a control experiment in which the original human choice rates were replaced with synthetic choice rates generated by an expected-value maximization model. Specifically, the synthetic choice rate for option B was set to 1 if option B had a greater expected value than option A, and 0 if it had a smaller expected value. In cases where the expected values of the two options were equal, the synthetic choice rate was set to 0.5.

Using the synthetic choice rate as the reward signal for RL (i.e., replacing $\mathbf{p}^B$ in Equation 1 with the synthetic choice rate), we find that the RL model quickly learns to generate CoT focused on computing and comparing expected values (see Figure 6b and Appendix D for details). Although risk aversion is no longer necessary to explain the synthetic data, the model still occasionally includes references to it in its CoT. This may be due to residual cues in the prompt suggesting that the task involves human risky choice (see Appendix A.2 and D.3 for details), even though the actual data had been replaced with synthetic data. Nonetheless, the RL model correctly identifies relevant cognitive mechanisms (e.g., rational choice theory and all-or-nothing thinking) as important for explaining the synthetic data. These results suggest that RL is capable of adapting its reasoning strategies to match the structure of the training data.

**Model control.** Additionally, we replicated the main experiment using a smaller and arguably weaker backbone LLM, Gemma-2-2B-Instruct (Team et al., 2024) (see Appendix E for details). The same LoRA configuration was applied, resulting in approximately 41.53M trainable parameters, corresponding to 1.56% of the total parameters in the 2B model. The Gemma model was fine-tuned using SFT, Centaur-style SFT, and RL on the same training set as in the main experiment and evaluated on the same held-out test set.

We observe several noteworthy findings. First, RL applied to the Gemma model performs significantly worse than both SFT and Centaur-style SFT in reducing MSE on the test set (see Figure 12 in Appendix E). This is a pattern that contrasts with the learning curves obtained using the stronger Qwen model. Second, analysis of the CoT generated during RL training reveals that the Gemma model fails to compute and compare the expected values of the risky options, a mechanism identified as critical for explaining human behavior in the main experiment (see Appendix E). This absence of key cognitive mechanisms in its CoT likely contributes to the weaker performance of RL on smaller models. Finally, even when RL underperforms, both SFT and Centaur-style SFT are still able to predict human behavior (albeit without explanation) at a level comparable to that of the stronger Qwen model. These suggest that RL and SFT exhibit different generalization patterns (Chu et al., 2025; Wang et al., 2025).

## 6  DISCUSSION

The emergence of reasoning LLMs presents new opportunities for addressing complex problems that traditional, non-reasoning LLMs struggle to solve. This improvement is likely due to the fact that CoT reasoning enhances the expressiveness of Transformer-based models (Li et al., 2024). Moreover, RL post-training has been shown to effectively elicit appropriate CoTs from backbone LLMs (Shao et al., 2024; Yue et al., 2025). This process can be seen as analogous to learning to reason like a psychologist: given sufficient capacity, the backbone LLM implicitly encodes a range of psychological theories about human risky choice, and RL serves to surface the most relevant theoretical representations through the generation of CoT reasoning. As a result, we trained an LLM-based psychological theorist capable of verbalizing relevant cognitive mechanisms for explaining and predicting human risky choices.

**RLVR, RLHF, and RLAIF.** Our work shows that applying reinforcement learning with verifiable or outcome-based rewards (RLVR) directly to human risky choices can elicit CoTs that explain those decisions *at scale*. Compared to SFT methods, RL methods have a distinct advantage: the resulting CoTs serve as verbal theories of human choices albeit at the cost of higher computational demands. An alternative to RLVR in the LLM setting is reinforcement learning from human feedback (RLHF; Lambert (2025); Ziegler et al. (2019)). Adapting RLHF to our context would require experts in risky decision-making to compare and evaluate LLM-generated explanations and predictions across different risky choice problems. For instance, scaling up the type of expert judgments illustrated in Figure 7. In practice, this would entail large-scale expert annotations to assess both the model's CoT explanations and its behavioral predictions, resources that are not currently available.

That said, we acknowledge that a scalable RL alternative may be possible by leveraging stronger LLMs as automated judges (e.g., role-playing expert psychologists), a strategy aligned with reinforcement learning from AI feedback (RLAIF; Bai et al. (2022); Lee et al. (2023)). Of course, scientists could also rely on black-box commercial RL training services (e.g., the RL fine-tuning API offered by OpenAI, n.d.). We chose not to use such services for comparison in the present work due to their lack of transparency, as the details of the underlying RL training are not disclosed.

**Training predictive models and then explaining their predictions.** As an alternative approach for obtaining interpretable theories of human cognition, researchers can first train predictive models and then apply modern explainability techniques (Agrawal et al., 2020). LLMs can also be used to strengthen the training of such predictive models; for example, synthetic data generated by LLMs has been shown to improve predictive accuracy in economic choice models (Shapira et al., 2024; 2025). This predict-then-explain pipeline is a promising alternative to our RL approach. However, note that the RL method simultaneously optimizes both predictions and explanations using reasoning LLMs. It also avoids the direct use of explainability techniques, which can sometimes be brittle for feature attributions (Bilodeau et al., 2024).

In principle, both approaches aim to uncover explainable and interpretable signals by improving behavioral predictions. The primary difference is that traditional explainability relies on features proposed by human researchers, whereas the RL approach automatically explores features or hypotheses that can explain human data using LLMs. LLM-proposed hypotheses are more automated but depend heavily on the base LLM's capacity to generate meaningful insights. Human-proposed hypotheses, while slower and less automated, may in some cases be more creative than those generated by LLMs.

**Limitations and future research.** A key limitation of developing cognitive models through CoT lies in the elicitation hypothesis of RL post-training (Lambert, 2025; Yue et al., 2025). If this hypothesis holds, RL is unlikely to generate entirely novel theories or cognitive mechanisms. For instance, imagine an LLM pretrained in a world prior to the development of Kahneman and Tversky's prospect theory (Kahneman & Tversky, 2013). RL alone would likely not discover this theory during post-training (see Appendix F for more discussion). However, in principle, RL can draw on theoretical frameworks beyond psychology to explain psychological phenomena. For example, it may learn to apply concepts from physics or computer science to human behavior, provided such ideas are already embedded in the backbone LLM. Future research could explore how RL facilitates the integration of knowledge across traditional disciplinary boundaries to generate novel explanatory frameworks.

Our findings also suggest that RL and SFT exhibit different generalization patterns. Further research is needed to systematically investigate these differences and understand the conditions under which each method performs the best. An additional open question is when and how to effectively combine RL and SFT to produce more robust and interpretable cognitive models through the CoT of LLMs.

**Conclusion.** We introduced a method for developing interpretable cognitive models using LLMs by eliciting CoT reasoning through RL post-training, using human risky choice as a case study. Our results show that RL post-training can generate meaningful CoTs that adapt to the structure of the training data, though the effectiveness of this approach strongly depends on the capabilities of the backbone LLM. We believe this method is highly generalizable and holds promise for applications in other domains of cognitive modeling.

## ETHICS STATEMENT

Our human experiment (reported in Appendix C.1) was approved by the Princeton University Institutional Review Board (IRB #10859: 'Computational Cognitive Science'). All participants provided informed consent prior to the experiment. Participants were asked to evaluate model completions on risky choice problems. We do not anticipate any additional ethical concerns.

## REPRODUCIBILITY STATEMENT

We provided complete descriptions of the training methods and associated hyperparameter values in the paper. All model training was conducted using standard Python packages from Hugging Face and vLLM.

## ACKNOWLEDGMENTS

JQZ and TLG acknowledge support from the NOMIS Foundation and computing resources provided by Princeton Language and Intelligence. HX acknowledges support from the OpenAI Researcher Access Program. RCW acknowledges funding from a SCIALOG Award (#29079) from the Research Corporation for Scientific Advancement. HX and RCW also acknowledge computing support in part through research cyberinfrastructure resources and services provided by the Partnership for an Advanced Computing Environment at the Georgia Institute of Technology, USA. DA and TLG were supported by ONR MURI N00014-24-1-2748.

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

## A PROMPTS

### A.1 PROMPTS FOR THE SFT AND CENTAUR-STYLE SFT MODELS

> **User:** You are a knowledgeable and insightful psychological theorist, skilled in analyzing human behavior, cognition, and decision-making. You will be shown two options, A and B. Your task is to estimate the proportion of people who will choose each option. Please only provide your final estimates in JSON format, ensuring that: "option_A" represents the percentage of people choosing Option A; "option_B" represents the percentage of people choosing Option B; The values must be numbers between 0 and 100 (inclusive); The sum of "option_A" and "option_B" must equal 100.

### A.2 PROMPTS FOR THE RL MODEL

> **User:** You are a knowledgeable and insightful psychological theorist, skilled in analyzing human behavior, cognition, and decision-making. You will be shown two options, A and B. Your task is to estimate the proportion of people who will choose each option. First, explain your reasoning step by step. Then, provide your final estimates in JSON format, ensuring that: "option_A" represents the percentage of people choosing Option A; "option_B" represents the percentage of people choosing Option B; The values must be numbers between 0 and 100 (inclusive); The sum of "option_A" and "option_B" must equal 100.

### A.3 EXAMPLE PROMPTS FOR RISKY OPTIONS

Risky choices from the `choices13k` dataset are described in natural language using the following format:

> Option A offers (1) a 50.0% chance to win $2.0; (2) a 50.0% chance to win $0.0.
>
> Option B offers a 100.0% chance to win $1.0.

## B COGNITIVE MECHANISMS IDENTIFIED BY QWEN-2.5-7B-INSTRUCT AFTER RL TRAINING

Based on the CoT reasoning generated by the RL models, we identified a set of cognitive mechanisms that the model appeared to find useful for predicting human risky choices. To assist with summarization, we used GPT-4.1 with the following prompt (the temperature for GPT-4.1 was fixed at 0):

> **User:** Read the following thought atom and return a JSON list of standard psychological effects or cognitive biases that are present. Use only the most relevant terms from established psychological concepts (e.g., "Expected Value", "Loss Aversion", "Risk Aversion", etc.). Return only a JSON list like ["Effect1", "Effect2", ...]. No explanation or extra text.

Cognitive mechanisms that appeared in at least 10 risky-choice problems are summarized in Table 1. Other mechanisms identified in the CoT reasoning are described below: *Social Proof, Reference Point, Reference Dependence, Behavioral Biases, Rational Choice Theory, Majority Effect, Gambler's Fallacy, Anchoring Effect, Risk Neutrality, Heuristic, Present Bias, Perceived Utility, Risk Premium, Representativeness Heuristic, Regret Aversion, Perceived Value, Heuristic Processing, Range Effect, Assumption Bias, Attractiveness Effect, Risk Assessment, Underweighting of Large Probabilities, Simplicity Preference, Perceived Risk, Simplicity Bias, Assumption of Uniformity, Diminishing Sensitivity, Variance Aversion, Diversification Bias, Variance Preference, Cognitive Biases, Mental Accounting, Endowment Effect, Neglect of Expected Value, Heuristic Bias, Overestimation of Small Probabilities, Heuristic Simplification, Sure-Thing Principle, Frequency Illusion, Psychological Biases, Pessimism Bias, Immediate Gratification, Preference for Variety, Ambiguity Effect, Overweighting of Gains, Immediate Reward Bias, Diversification Effect, Overestimation of Rare Events, Contrast Effect, Regret Theory, Multiple Gains Effect, Equally Likely Heuristic, Approximation Bias, Overweighting Small Probabilities, Positivity Effect, Salience of Losses, Affect*

Table 1: Frequency of cognitive mechanisms identified in the CoT reasoning generated by the RL model (final checkpoint). Only mechanisms that appeared in at least 10 risky-choice problems are reported.

| Cognitive mechanism | Count |
| --- | --- |
| Expected value | 6495 |
| Risk aversion | 5210 |
| Loss aversion | 1676 |
| Certainty effect | 1453 |
| Risk seeking | 710 |
| Probability weighting | 470 |
| Framing effect | 278 |
| Prospect theory | 113 |
| Ambiguity aversion | 99 |
| Risk perception | 76 |
| Overweighting of small probabilities | 64 |
| Risk tolerance | 64 |
| Heuristics | 51 |
| Attraction effect | 50 |
| Assumption of equiprobability | 49 |
| Equiprobability bias | 41 |
| Diminishing marginal utility | 38 |
| Optimism bias | 37 |
| Risk preference | 34 |
| Possibility effect | 30 |
| Utility theory | 23 |
| Expected utility theory | 19 |
| Availability heuristics | 19 |
| Anchoring | 18 |
| Variance effect | 15 |
| Bandwagon effect | 14 |
| Salience bias | 13 |
| Diversification | 12 |
| Probability overestimation | 12 |
| Assumption of equal probability | 12 |
| Complexity aversion | 11 |
| Expected utility | 10 |
| Expected value neglect | 10 |

*Heuristic, Windfall Effect, Underweighting of Losses, Magnitude Effect, Majority Influence, Frequency Effect, Nonlinear Utility, Risk-Reward Tradeoff, Sure Thing Effect, Heuristic Biases, Overconfidence Effect, Preference for Simplicity, Law of Small Numbers, Attractiveness Heuristic, Denomination Effect, Cognitive Ease, Estimation Bias, Value Perception, Allais Paradox, Probability Perception, Utility of Money, Psychophysical Numbing, Hope Effect, Assumption of Symmetry, Simplicity Effect, Salience, Decision Heuristics, Probability Matching, Weber-Fechner Law, Satiation, Overconfidence Bias, Assumption of Probability Distribution, Assumption Heuristic, Small Wins Effect, Reward Seeking, Intuition, Neglect of Probability Weighting, Lottery Effect, Time Preference, Range Seeking, Impact Bias, Positive Outcome Bias, Reference Points, Fairness Bias, Risk Seeking in Losses, Loss Underestimation, Heuristic Substitution, Behavioral Economics, Temporal Discounting, Risk-Reward Ratio, Lump Sum Effect, Overestimation, Gambling Fallacy, Variety Seeking, Framing, Variability Effect, Anchoring Bias, Risk-Seeking, Heuristic Reasoning, Conservatism Bias, Delay Discounting, Undervaluation of Large Losses, Overweighting of Large Gains, Reward Sensitivity, Minimization of Differences, Reference Point Effect, Uncertainty Aversion, Simplification, Underweighting of Small Probabilities, Immediate Reward Preference, Overestimation Effect, Motivational Salience, Overweighting of Outcomes, All-or-Nothing Thinking, Simplification Bias, Equal Probability Bias, Segregation of Gains, Choice Overload, Social Norms, Majority Illusion, Near Miss Effect, Psychological Value of Money, Frequency Bias, Normalization, Emotional Valence, Risk-seeking Behavior.*

## C  ADDITIONAL ANALYSES OF CoT

### C.1  HUMAN EVALUATION OF CoT

To obtain a preliminary human evaluation of CoT reasoning quality, we conducted the following experiment.

**Participants.** We recruited 20 participants via a social media post targeting local universities, inviting volunteers to evaluate completions generated by LLMs. The sample included 12 graduate students, 3 postdoctoral researchers, 3 faculty members, and 2 undergraduate students. Participants represented a range of academic disciplines: 10 from Psychology or Cognitive Science, 7 from Computer Science or Artificial Intelligence, and 3 from Business or Economics. On average, participants reported a self-assessed knowledge of human risky decision-making of 3.5 out of 5. Participants did not receive any monetary payment, and the study lasted up to 30 minutes. All experimental sessions were conducted in May 2025. Informed consent was obtained from all participants (Princeton University IRB number 10859: 'Computational Cognitive Science')

**Procedures.** Each participant completed 10 evaluation trials. On each trial, a risky choice problem was randomly sampled from the held-out test set. The corresponding CoT completions generated by the backbone and RL models were presented side by side beneath the problem statement (see Figure 7 for an example). To ensure that evaluations focused on reasoning quality rather than predictive accuracy, we removed the final choice predictions from the completions. Participants were instructed to select which CoT they found more reasonable by making a binary choice. After each choice, they also reported their confidence using a slider ranging from 0 (least confident) to 100 (most confident). To prevent bias, the two completions were anonymized as Model A and Model B, and their left–right order was randomized across trials.

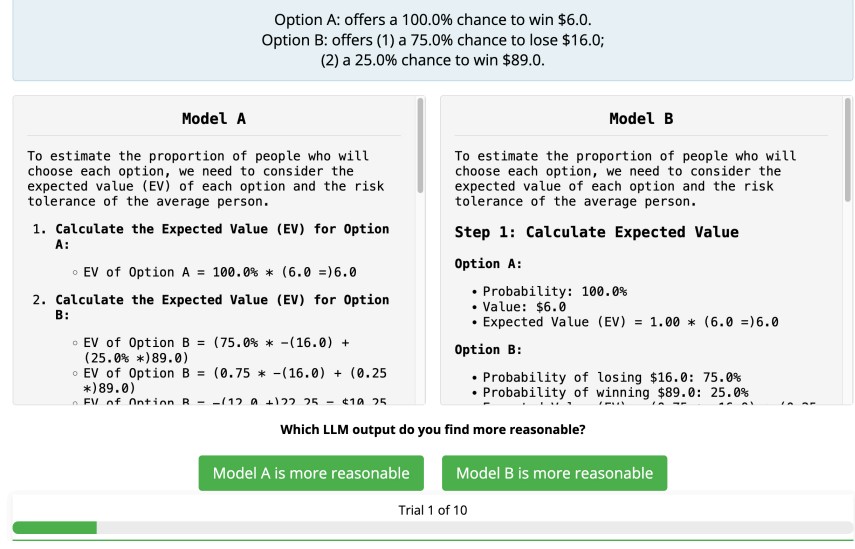

Figure 7: Screenshot of a representative trial from the human evaluation experiment. Participants were shown a risky choice problem followed by two anonymized CoT completions (Model A and Model B) and asked to select the more reasonable explanation.

**Results.** On average, CoTs generated by the RL model were selected as more reasonable in 61.5% of trials ($SE = 5.2\%$) above the chance level of 50% at 0.05 significance level ($t(19) = 2.19$, $p < .05$). This suggests that participants found the reasoning produced by the RL model to be more persuasive or coherent than that of the backbone model. While this represents a useful initial step in soliciting expert opinions from researchers in relevant fields, scaling human evaluations to the whole dataset remains a significant challenge.

## C.2 USING GPT-4.1 TO JUDGE COT

Due to the large volume of CoTs generated across multiple model checkpoints and risky choice problems, conducting comprehensive human evaluations is constrained by time and labor demands. To address this limitation, we piloted an alternative approach using OpenAI's GPT-4.1 to simulate expert evaluations. Specifically, GPT-4.1 was instructed to role-play as an expert in judgment and decision-making and provide quality ratings of the CoTs. The temperature for GPT-4.1 was fixed at 0. The model was prompted with the following instructions:

> **System:** You are an expert in judgment and decision-making.
>
> **User:** As an expert in judgment and decision-making, please evaluate the reasoning and prediction of the following question. Provide a single integer score from 0 to 100 based on the quality of the completion.

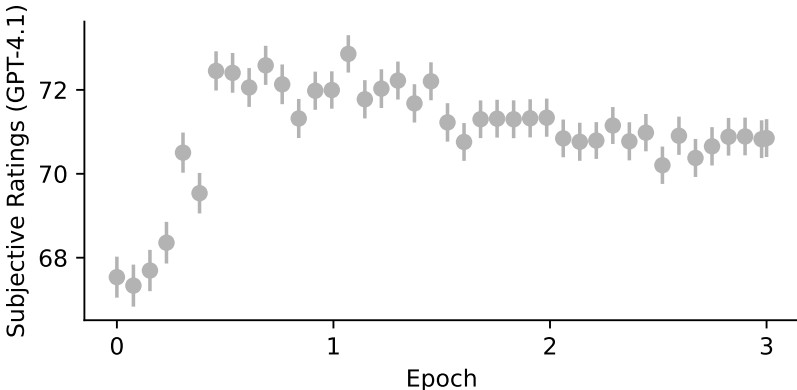

Figure 8: GPT-4.1 was prompted to act as an expert evaluator, rating the RL model's completions for risky-choice problems in the test set. Error bars represent $\pm 1$ standard error across choice problems.

As shown in Figure 8, GPT-4.1's ratings of the RL model's completions show a modest initial increase, though the magnitude of change is relatively small given the full rating scale of 0 to 100. The scores then stabilize for several training steps before exhibiting a slight decline toward the end. This pattern suggests that the quality of the CoT reasoning largely plateaued after the first epoch of RL training.

## C.3 SWAPPING COT BETWEEN BACKBONE AND RL MODELS

To examine the potential causal relationship between CoT reasoning and final predictions of human choice proportions, we conducted the following analysis. First, we evaluated both the original backbone LLM (i.e., Qwen-2.5-7B-Instruct, or checkpoint 0) and its RL post-trained version (i.e., the final RL checkpoint) on the test set, recording their generated CoTs and final predictions (see the first row of Table 2). Next, we removed the final JSON prediction from each model's completion and performed a CoT-swapping procedure: the backbone LLM was prompted with the RL model's CoT and asked to continue the generation, while the RL model was prompted with the backbone LLM's CoT and similarly asked to produce a final prediction (see the second row of Table 2).

We find that using the RL model's CoT significantly reduces the backbone model's prediction error, as measured by MSE ($t(2596) = -18.11, p < .01$). Conversely, using the backbone model's CoT significantly increases the RL model's prediction error ($t(2597) = 20.65, p < .01$). These results suggest that the RL model generates higher-quality CoTs that contribute more effectively to accurate final predictions than those produced by the backbone model.

Table 2: Prediction errors for the backbone LLM and its RL post-trained counterpart. Values represent mean squared error on the test set, with standard errors in parentheses. Original CoT indicates that the model used its own CoT reasoning for prediction, whereas Swapped CoT refers to predictions made using the CoT generated by the other model.

|  | Qwen-2.5-7B-Instruct (backbone) | Qwen-2.5-7B-GRPO-c13k (RL) |
|---|---|---|
| Original CoT | .0694 (.0025) | .0148 (.0006) |
| Swapped CoT | .0212 (.0008) | .0695 (.0025) |

## C.4 TESTING THE CoTs GENERATED BY SFT AND CENTAUR MODELS

When an LLM is SFT-trained on risky choices, it may also learn the behavioral policy reflected in the risky-choice data (Betley et al., 2025). For example, if the choice data are risk-seeking, the finetuned LLM will exhibit more risk-seeking behavior as well. Therefore, to further test whether the CoTs elicited using RL are more informative than those produced via SFT, we also elicited CoTs from SFT models and evaluate whether these SFT-generated CoTs are effective at predicting human risky choices. Specifically, we prompted both the SFT and Centaur models (i.e., the final SFT and Centaur checkpoints) with the RL prompt that encourages step-by-step reasoning before providing a behavioral prediction in JSON format. We then removed the final JSON prediction from the model completions and fed the remaining CoT, without the JSON output, to the backbone LLM – a procedure similar to the CoT-swapping experiment reported in Table 2.

We find that neither SFT-generated nor Centaur-generated CoTs improve the backbone model's predictions of human risky choices. The MSE of the backbone LLM's predictions using SFT-generated CoTs is $M = 0.0785$ ($SE = 0.0023$), and the MSE using Centaur-generated CoTs is $M = 0.0728$ ($SE = 0.0019$). Both values are comparable to the performance of the backbone LLM using its original CoT (see the top-left cell of Table 2). These results suggest that although SFT and Centaur models can achieve comparable behavioral predictions without CoTs, the CoTs generated by these models do not help improve the predictions of the original LLM. This highlights the particular need for RL post-training, beyond SFT and Centaur-style SFT, to produce more effective CoTs.

Finally, we examined the evolution of SFT-generated and Centaur-generated CoTs over the course of training. Similar to the analysis in Figure 6, where we used GPT-4.1 to summarize CoTs generated by the RL models, we also prompted GPT-4.1 with the SFT-generated and Centaur-generated CoTs (see Figure 9). Compared to the CoTs elicited through RL training in Figure 6, both SFT and Centaur-style SFT (neither of which are explicitly trained to produce CoTs) showed no noticeable changes in CoT frequencies across training epochs. We do observe an upward trend in the usage of "expected value" in the SFT-generated CoTs (Figure 9a), whereas its usage decreases in the Centaur-generated CoTs (Figure 9b). This suggests that even though SFT and Centaur-style SFT yield similar behavioral predictions, the CoTs they generate can differ.

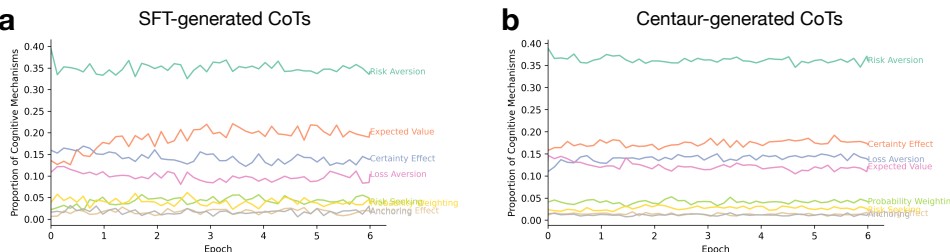

Figure 9: Evolution of Chain-of-Thoughts (CoTs) generated by the SFT and Centaur-style SFT models over training epochs. (a) SFT-generated CoTs. (b) Centaur-generated CoTs.

# D DATA CONTROL EXPERIMENT

## D.1 LEARNING CURVES

In the data control experiment, we retained the same backbone LLM, Qwen-2.5-7B-Instruct, but replaced the human data with synthetically generated data from an expected-value maximization model. We then performed RL post-training on this synthetic dataset using the same set of hyperparameters as in the main experiment. The RL learning curve shows a gradual improvement in model performance over training steps (see Figure 10).

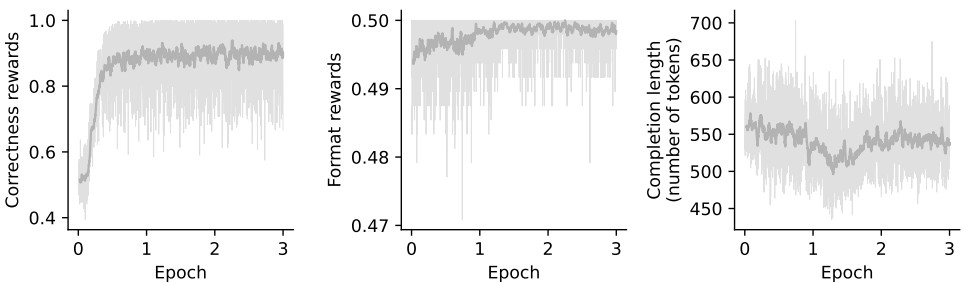

Figure 10: **Data control experiment.** Learning curves for the RL model. Backbone LLM is Qwen-2.5-7B-Instruct. (*Left*) Correctness reward, defined as one minus the absolute difference between model predictions and human choice proportions. (*Middle*) Format reward, based on the structure and position of the model's JSON output. (*Right*) Completion length, measured by the number of generated tokens.

## D.2 COGNITIVE MECHANISMS IDENTIFIED BY QWEN-2.5-7B-INSTRUCT DURING THE DATA CONTROL EXPERIMENT

We replicated the CoT analysis presented in Appendix B for the RL model trained on the synthetic choice dataset in the control experiment. The evolution of the top eight cognitive mechanisms identified in the model's CoT reasoning across training epochs is shown in Figure 6b. Given that the synthetic choice rates were generated by an expected-value maximizer model, the expected value computation emerges as the dominant mechanism in the CoT.

The full list of cognitive mechanisms, ordered by frequency of use, is presented below: *Expected Value, Risk Aversion, Rational Choice Theory, All-or-Nothing Thinking, Black-and-White Thinking, Certainty Effect, Herd Behavior, Bandwagon Effect, Consensus Bias, Expected Utility Theory, Zero-Sum Bias, Overconfidence Effect, Loss Aversion, Risk Seeking, Probability Neglect, Majority Influence, False Consensus Effect, Dominance Heuristic, Indifference, Overconfidence Bias, Overweighting of Small Probabilities, Indifference Principle, Binary Bias, Social Proof, Majority Effect, Equiprobability Bias, Contrast Effect, Optimism Bias, Risk Neutrality, Indifference Point, Expected Utility, Ambiguity Aversion, Equity Heuristic, Normative Decision Theory*

## D.3 ADDITIONAL DATA CONTROL EXPERIMENTS

To further verify that the CoTs elicited from LLMs through RL training adapt to the structure of the training dataset, we conducted two additional control experiments using (i) a Random Dataset, consisting of randomly generated choice rates (i.e., samples from $U[0, 1]$), and (ii) a Complexity Aversion Dataset, in which choice rates reflect only complexity aversion. Specifically for (ii), we define complexity based on the number of outcomes in each gamble. For example, if Option A had one outcome and Option B had two, the synthetic data specified that $2/3$ of people would prefer Option A (i.e., $\frac{\text{num\_outcomes\_B}}{\text{num\_outcomes\_A}+\text{num\_outcomes\_B}}$), and $1/3$ would choose Option B (i.e., $\frac{\text{num\_outcomes\_A}}{\text{num\_outcomes\_A}+\text{num\_outcomes\_B}}$) regardless of the underlying probabilities or payoff values. In essence, the only decision rule was to probabilistically prefer the option with fewer outcomes.

We then trained the Qwen-2.5-7B-Instruct model using the same RL setup in the main text. When trained on the Random Dataset, the model collapsed to consistently predicting 50% choice rates, and its CoT generations became nonsensical, often degenerating into multilingual gibberish. In contrast, when trained on the Complexity Aversion Dataset, the learning curve showed a steady increase in correctness reward toward 0.97 (with 1 as the maximum), indicating that the model has successfully learned to predict complexity-averse behavior. Moreover, the resulting CoTs generated by the RL model more frequently referenced concepts such as "simplicity" and "complexity" (e.g., "Complexity: Option B is significantly more complex and involves a higher number of possible outcomes. People often avoid options with too many uncertainties and choices, which can lead to decision fatigue and heuristic shortcuts.").

However, as shown in Figure 11a, the CoTs did not exclusively focus on complexity, likely due to residual influence from the original prompt, which framed the task as explaining human risky choice. This phenomenon mirrors our earlier data control experiment in which synthetic data were generated by an expected-value maximization model. However, the relative rank of thoughts that mention complexity aversion steadily increases over the course of RL training, rising from around 16th place to around 6th place (see Figure 11b). Taken together, these additional control experiments further support the flexibility and robustness of our RLVR-based approach in eliciting sensible CoT explanations from LLMs that adapt to the structure of the training data.

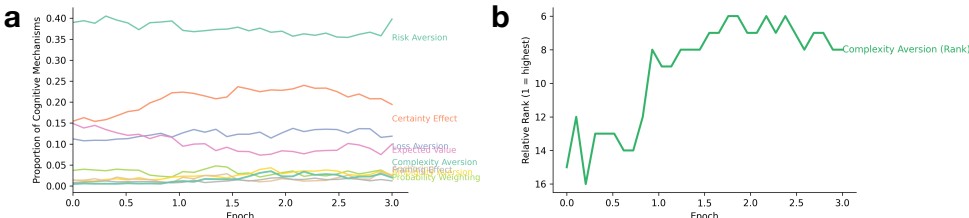

Figure 11: Proportions of cognitive mechanisms identified in the RL models' thoughts across training epochs in the additional data control experiment, where the synthetic dataset reflects complexity-aversion behavior. **(a)** The eight most frequently used mechanisms are shown. **(b)** The relative rank of complexity-aversion usage in the CoTs increases over the course of RL training.

## E    MODEL CONTROL EXPERIMENT

### E.1    LEARNING CURVES

In the model control experiment, we replaced the original Qwen-2.5-7B-Instruct model with the smaller Gemma-2-2B-Instruct model, while keeping the human risky choice data as the target for prediction. We find that SFT and Centaur-style SFT achieve comparable levels of predictive accuracy (see Figure 12). At the final checkpoint, the MSE on the test set is $M = 0.0162, SE = 0.0006$ for SFT and $M = 0.0163, SE = 0.0007$ for Centaur-style SFT, with no statistically significant difference between them ($t(2924) = -0.13, p = 0.90$). In contrast, RL yields significantly higher MSE at its final checkpoint ($M = 0.0526, SE = 0.0016$), indicating poorer predictive performance.

### E.2    COGNITIVE MECHANISMS IDENTIFIED BY GEMMA-2-2B-INSTRUCT

As before, we replicated the CoT analysis (see Appendix B) for the RL model trained using the Gemma-2-2B-Instruct backbone. This model performs poorly in predicting human choices compared to both SFT-based methods applied to the same LLM and RL applied to the stronger Qwen model. The distribution of identified cognitive mechanisms in the CoT provides insight into this performance gap (see Figure 14). Notably, the CoT lacks arguably the most critical component: expected value calculation and comparison. The absence or infrequent usage of this key cognitive mechanism likely contributes to the RL model's failure to accurately capture human risky choice.

Additional identified mechanisms are listed below in order of frequency: *Ambiguity Aversion, Risk Tolerance, Probability Weighting, Anchoring, Subjective Probability, Anchoring Bias, Probability*

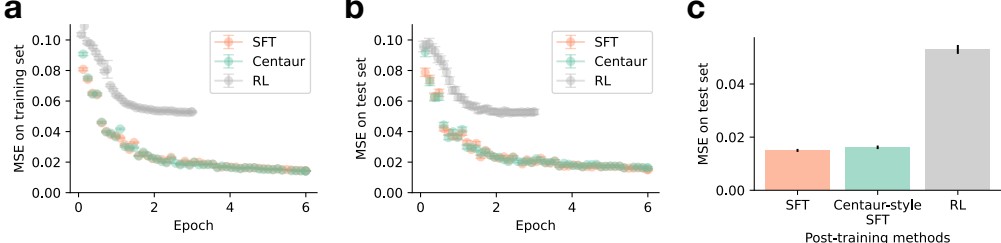

Figure 12: **Model control experiment.** Learning curves on the **(a)** training and **(b)** test sets. Backbone LLM is Gemma-2-2B-Instruct. The horizontal axes indicate training epochs, while the vertical axes represent mean squared error (MSE) evaluated on the corresponding dataset. The three post-training strategies compared are supervised fine-tuning (SFT, red), Centaur-style SFT (green), and reinforcement learning using Group Relative Policy Optimization (RL, grey). **(c)** MSE on the test set at the final checkpoint of each post-training method. Error bars represent ±1 standard error across risky-choice problems.

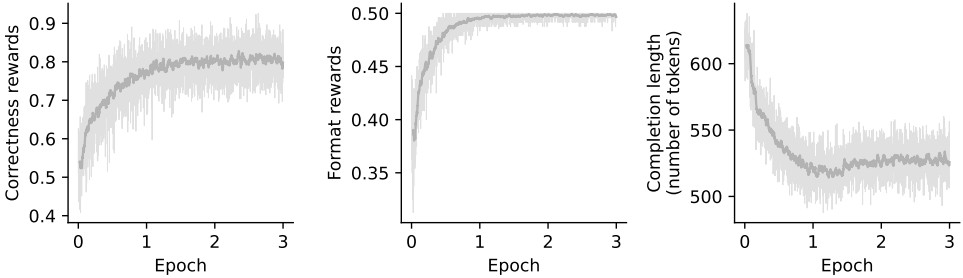

Figure 13: **Model control experiment.** Learning curves for the RL model. Backbone LLM is Gemma-2-2B-Instruct. (*Left*) Correctness reward, defined as one minus the absolute difference between model predictions and human choice proportions. (*Middle*) Format reward, based on the structure and position of the model's JSON output. (*Right*) Completion length, measured by the number of generated tokens.

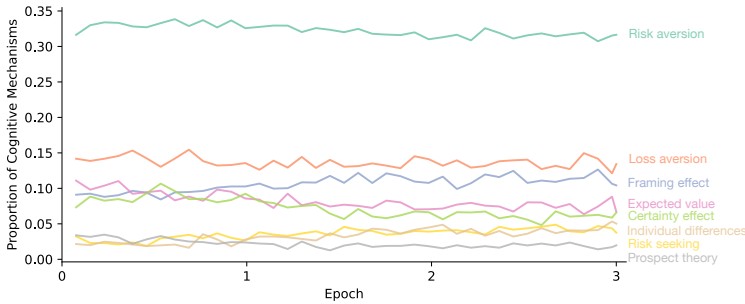

Figure 14: **Model control experiment.** Proportions of cognitive mechanisms identified in the RL models' thoughts across training epochs. Backbone LLM is Gemma-2-2B-Instruct. The top 8 most frequently used mechanisms are displayed.

*Neglect, Anchoring Effect, Status Quo Bias, Mental Accounting, Affect Heuristic, Uncertainty Aversion, Immediate Gratification, Social Proof, Endowment Effect, Optimism Bias, Confirmation Bias, Present Bias, Heuristics, Social Influence, Availability Heuristic, Attraction Effect, Context Effect, Opportunity Cost, Possibility Effect, Regret Aversion, Overconfidence Effect, Gambler's Fallacy, Hedonic Adaptation, Personality Traits, Lottery Effect, Reward Sensitivity, Overweighting of Small Probabilities, Choice Overload, Emotional Reasoning, Impulsivity, Comparative Judgment, Group-*

*think, Reward Seeking, Cognitive Bias, Cognitive Dissonance, Stress Influence on Decision Making, Cognitive Complexity, Desire for Gain, Pessimism Bias, Relative Value, Cognitive Biases, Information Overload, Mood Congruent Bias, Expected Utility Theory, Immediate Reward Bias, Probability Perception, Subjective Value, Illusion of Control, Perceived Probability, Desirability Bias, Heuristic, Cognitive Load, Bounded Rationality, Sunk Cost Fallacy, Ambiguity Effect, Positivity Bias, Comparative Evaluation, Cognitive Appraisal, Perceived Value, Normative Social Influence, Expected Value Neglect, Herd Behavior, Context Dependence, Incentive Salience, Risk Perception, Intuition, Subjective Utility, Salience Bias, Subjectivity, Neglect of Probability, Personal Experience Effect, Magnitude Effect, Motivation, Overestimation of Small Probabilities, Hope, Hot-Hand Fallacy, Information Asymmetry, Probability Overestimation, Value of Money, Mental Fatigue, Complexity Aversion, Uncertainty Effect, Reference Dependence, Time Pressure, Perceived Predictability, Fear of Missing Out.*

## F  ADDITIONAL DISCUSSION

**Failures of RL.** During our experiments, we also observed several failures associated with GRPO training. Most notably, applying the original GRPO algorithm (Shao et al., 2024) to the `choices13k` dataset led to mode collapse. By mode collapse, we refer to the phenomenon in which the fine-tuned LLMs converge to generating a single, repetitive reasoning chain that simply computes the expected values of both options and consistently outputs a final prediction of 50% for each option across all choice problems. Note that the `choices13k` dataset exhibits a modal human choice proportion of 50%, which may have contributed to this degeneracy.

We experimented with several strategies to prevent the model from collapsing to this default prediction. Specifically, we tested alternative reward functions, including one minus the mean squared error, negative cross-entropy loss, and the addition of diversity bonuses incentivizing diverse predictions. However, none of these modifications were effective in mitigating mode collapse in our setting. The issue was eventually addressed by removing advantage normalization in GRPO, as recommended by Liu et al. (2025). That is, dividing the centered reward by the standard deviation of group rewards introduces a bias toward question-level difficulty: risky-choice problems with lower standard deviations (i.e., those that are either too easy or too difficult) disproportionately influence policy updates by receiving higher weights. By omitting this normalization step, we were able to recover the main RL results reported above.

After removing normalization, the stability of RL training improved substantially. We then evaluated three reward functions under GRPO without normalization: (i) $1 -$ absolute prediction error, (ii) $1 -$ squared error, and (iii) negative cross-entropy loss. Among these, reward function (i) yielded the most stable RL training, characterized by smoother KL trajectories and smaller reward standard deviations compared to (ii) and (iii). We suspect this advantage arises because (i) aligns more closely with GRPO's centered reward design.

**The elicitation hypothesis.** A prominent hypothesis about RL post-training is that it primarily increases the probability of generating correct outputs by eliciting behaviors already latent in the pretrained LLM (Lambert, 2025; Yue et al., 2025). In this view, RL post-training does not teach new capabilities but instead amplifies and selects from pre-existing knowledge. Our model control results support this hypothesis: weaker LLMs — particularly those unable to elicit the expected value mechanism for risky choice — struggle to explain human behavior effectively when trained with RL. These results suggest that the success of RL highly depends on the capacity of the backbone LLM.

## G  IMPLEMENTATION DETAILS

**7B models.** RL training was conducted over 3 epochs using the training set, distributed across $4\times$H100 GPUs for a total runtime of 80 hours. For SFT and Centaur-style SFT trainings, each model was trained for 6 epochs on the training set, using a single A100 GPU with an approximate runtime of 5 hours per training session.

Inference across all checkpoints on the complete `choices13k` dataset took approximately 5 hours on a single A100 GPU for the SFT and Centaur-style SFT models using vLLM (Kwon et al., 2023), whereas the RL models required roughly 30 hours under the same conditions.

**2B models.** RL training was conducted over 3 epochs using the training set, distributed across $4\times$H100 GPUs for a total runtime of 40 hours. For SFT and Centaur-style SFT trainings, each model was trained for 6 epochs on the training set, using a single A100 GPU with an approximate runtime of 3 hours per training session.

Inference across all checkpoints on the complete `choices13k` dataset took approximately 3 hours on a single A100 GPU for the SFT and Centaur-style SFT models using vLLM (Kwon et al., 2023), whereas the RL models required roughly 20 hours under the same conditions.

**Pilot and control experiments.** We also conducted several RL runs for pilot and control experiments, which together accounted for approximately 2,500 H100 GPU hours.

