# OpenReview forum: "Using Reinforcement Learning to Train Large Language Models to Explain Human Decisions"
_ICLR.cc/2026/Conference — ICLR 2026 Poster_

### Official Review · Reviewer_ABiG · 2025-10-17

**Soundness:** 3
**Presentation:** 3
**Contribution:** 3
**Rating:** 6
**Confidence:** 3

**Summary:**

The paper studies the capabilities of LLMs in explaining human decision-making in risky choice scenarios. They compare three methods: a standard SFT, a variation of SFT used in the Centaur model (where only human data tokens affect the loss), and an RL-based approach where the GRPO algorithm is applied to model-generated candidates. The core idea is that CoT can be used to derive explanations for the predicted choices. The experimental results indicate that all three methods achieve comparable performance in terms of predictive power. The paper then analyzes the CoT streams of the RL-based method and utilizes them to characterize cognitive mechanisms in the risky-choice task. Finally, controlled experiments show that the RL-based approach can also learn explanations for synthetic expected-value-maximization choice data, and that the weaker backbone model leads to weaker performance on the RL method compared to the SFT alternatives.

**Strengths:**

1. The paper is well written and addresses an important and timely topic at the intersection of AI, behavioral economics, and cognitive sciences, which is likely to be of interest for a large share of the ML community.

2. The use of CoT for explaining human choice is an elegant idea with potential applications that extend well beyond the scope of this paper, making it a promising direction to pursue.

3. The experiments are comprehensive and rigorous, leading to important insights on the capabilities of LLMs not only in predicting but also in explaining human choice and behavior.

**Weaknesses:**

While the paper offers a thorough examination of explaining human choices through various forms of LLM fine-tuning, it overlooks some natural alternative approaches to this problem, both in the literature review and in the experimental baselines.

One such approach would be to train a predictive ML model as a baseline predictor and then apply standard explainability techniques to identify which features drive its predictive performance. LLMs could still play a role in this framework - for example, in feature engineering or in interpreting feature importance measures. While I acknowledge that such methods may fall somewhat outside the paper’s primary scope, I would nonetheless expect some discussion of them: where might they fall short, and under what conditions might they provide sufficiently meaningful explanations?

Related work that could be referenced here includes Marantz & Plonsky (2025), who study the prediction of human choices in textually described lotteries, and Shapira et al. (2024, 2025), who explore data generation for improving human choice prediction in repeated persuasion games. Although these studies focus more on prediction than explanation, they nevertheless offer methodological insights into understanding decision-making in risky environments. For instance, comparing the performance of different predictive models can itself reveal patterns about how decisions are made. It would be valuable for the paper to discuss whether similar insights can be derived using the proposed approach, or whether the two approaches are fundamentally orthogonal. Such a discussion would significantly strengthen the paper’s contribution.

**References**

Marantz, E., & Plonsky, O. (2025). Predicting Human Choice Between Textually Described Lotteries.

Shapira, E., Madmon, O., Apel, R., Tennenholtz, M., & Reichart, R. (2025). Human choice prediction in language-based persuasion games: Simulation-based off-policy evaluation.

Shapira, E., Madmon, O., Reichart, R., & Tennenholtz, M. (2024). Can llms replace economic choice prediction labs? the case of language-based persuasion games.

**Questions:**

The paper is concerned with explaining human choice in binary choice tasks at the population level. I wonder if you have any thoughts/ideas about how the RL-based approach could be extended beyond such settings, for instance:
- predicted action is a high-dimensional object (e.g., free text) rather than a binary choice; and
- predicting behavior at the individual level rather than in the population level.

Additionally, I encourage the authors to share their opinion regarding the points raised in the 'weaknesses' section.

---

> ### Author Response · Authors · 2025-11-25
>
> >While the paper offers a thorough examination of explaining human choices through various forms of LLM fine-tuning, it overlooks some natural alternative approaches to this problem, both in the literature review and in the experimental baselines.
> One such approach would be to train a predictive ML model as a baseline predictor and then apply standard explainability techniques to identify which features drive its predictive performance. LLMs could still play a role in this framework - for example, in feature engineering or in interpreting feature importance measures. While I acknowledge that such methods may fall somewhat outside the paper’s primary scope, I would nonetheless expect some discussion of them: where might they fall short, and under what conditions might they provide sufficiently meaningful explanations?
> Related work that could be referenced here includes Marantz & Plonsky (2025), who study the prediction of human choices in textually described lotteries, and Shapira et al. (2024, 2025), who explore data generation for improving human choice prediction in repeated persuasion games. Although these studies focus more on prediction than explanation, they nevertheless offer methodological insights into understanding decision-making in risky environments. For instance, comparing the performance of different predictive models can itself reveal patterns about how decisions are made. It would be valuable for the paper to discuss whether similar insights can be derived using the proposed approach, or whether the two approaches are fundamentally orthogonal. Such a discussion would significantly strengthen the paper’s contribution.
>
> >Additionally, I encourage the authors to share their opinion regarding the points raised in the 'weaknesses' section.
>
> **Response:** Thank you for raising the potential connection with explainability work in machine learning. In typical ML workflows, one first trains a strong predictive model and then applies explainability techniques to understand why the model makes particular predictions. LLMs can also be used to generate synthetic data to improve the generalization of such predictive models. This procedure treats prediction and explanation as largely separate stages, with the LLM serving primarily a supportive role in strengthening the predictive model. In contrast, our RL approach leverages reasoning LLMs to simultaneously optimize for both predictions and explanations. We believe both approaches are valuable, each with its own strengths and limitations.
>
> In principle, both approaches aim to uncover explainable and interpretable signals from improving behavioral predictions. The primary difference is that traditional explainability relies on features proposed by human researchers, whereas the RL approach automatically explores features or hypotheses that can explain human data using LLMs. LLM-proposed hypotheses are more automated but depend heavily on the base LLM’s capacity to generate meaningful hypotheses. Human-proposed hypotheses, while slower and less automated, may in some cases be more creative than those generated by LLMs.
>
> To better contextualize our contribution relative to this broader landscape, we have now included a new Discussion section: “Training predictive models and then explaining their predictions.”

---

> > ### Author Response · Authors · 2025-11-25
> >
> > >The paper is concerned with explaining human choice in binary choice tasks at the population level. I wonder if you have any thoughts/ideas about how the RL-based approach could be extended beyond such settings, for instance:
> > predicted action is a high-dimensional object (e.g., free text) rather than a binary choice; and
> > predicting behavior at the individual level rather than in the population level.
> >
> > **Response:** Thank you for raising these interesting questions. Extending our RL-based approach to more complex scenarios would indeed require careful modifications to the reward function and training methodology, although the core ideas of RL remain applicable in these settings.
> >
> > For scenarios involving high-dimensional actions:
> > - N-Discrete Choices: When moving beyond a binary choice to an N-alternative choice problem, the reward function should be adapted to quantify the discrepancy between the model’s predicted choice distribution and the ground-truth distribution. For example, the reward could be defined using a distributional divergence metric.
> >
> > - Free-Text Generation: For tasks involving free-text (and thus high-dimensional) actions, an RLHF-style pipeline would likely be necessary. This would involve training an intermediate reward model to capture human preferences for the desired textual output, which would then provide a generalized reward score for RL training.
> >
> > For individual-level training:
> > - The key idea is to reveal individual differences through the training process, which requires adjustments to both prompts and reward design. For example, explicit individual identifiers (e.g., subject ID or other metadata) could be included in the prompt to distinguish individuals during training. Additionally, one may want to balance the contribution of different individuals to the pooled training by modifying the reward function, such as using curriculum learning or normalizing behavioral metrics, before assigning rewards.

---

> > > ### Comment · Reviewer_ABiG · 2025-11-25
> > >
> > > I appreciate the authors' detailed response and maintain my positive score.

---

### Official Review · Reviewer_Y1tw · 2025-10-20

**Soundness:** 2
**Presentation:** 2
**Contribution:** 1
**Rating:** 2
**Confidence:** 4

**Summary:**

This paper proposes to use RL to train language models to explain humans' cognitive behaviors. Compared to the previous approach that only focuses on the prediction accuracy, this paper prompts the language model to generate reasoning and prediction where the reasoning path can be used to explain human behaviors. The problem designs an outcome reward that checks whether the prediction is correct or not and uses GRPO to train the language model.

**Strengths:**

The problem of explaining human decisions is interesting and worth exploring.

**Weaknesses:**

I do not see the contribution of this paper. The outcome reward that compares prediction correctness is standard in RLVR, the GRPO is directly borrowed from literature, the "step-by-step" prompt is also directly borrowed from literature. In that sense, I do not see anything new or unique that is proposed by this paper.

Additionally, this paper focuses on the interpretability of human decisions, however, I do not see any special design for this purpose. Specifically, the paper proposes an outcome reward that checks prediction correctness, which is no different from the way where we only care about the prediction accuracy. One potential solution is that the paper can design some process reward to help supervise the reasoning/interpretability path.

**Questions:**

Can the authors elaborate the unique design of this paper or special design for explaining human decisions?

---

> ### Author Response · Authors · 2025-11-25
>
> >I do not see the contribution of this paper. The outcome reward that compares prediction correctness is standard in RLVR, the GRPO is directly borrowed from literature, the "step-by-step" prompt is also directly borrowed from literature. In that sense, I do not see anything new or unique that is proposed by this paper.
> Additionally, this paper focuses on the interpretability of human decisions, however, I do not see any special design for this purpose. Specifically, the paper proposes an outcome reward that checks prediction correctness, which is no different from the way where we only care about the prediction accuracy. One potential solution is that the paper can design some process reward to help supervise the reasoning/interpretability path.
>
> >Can the authors elaborate the unique design of this paper or special design for explaining human decisions?
>
> **Response:** We would like to clarify our contributions are intended to be to the “Applications to Neuroscience & Cognitive Science” area of ICLR (https://iclr.cc/Conferences/2026/CallForPapers). Traditional cognitive modeling has focused on improving predictive accuracy on human data. Models with better fits (or better predictive accuracy) are typically taken as evidence for the cognitive mechanisms they assume. In the era of LLMs, this conventional practice of cognitive modeling has scaled up; for example, the Centaur model directly fits human data described in natural language using Centaur-style SFT (Binz et al., 2025, Nature). However, the finetuned LLM (i.e., the Centaur model) is not directly interpretable, as we do not know what makes Centaur better at predicting human behavior.
>
> This motivates our work: we use RL to elicit interpretable CoTs from LLMs so that cognitive scientists can examine these CoTs as candidate verbal theories of human behavior, while also evaluating the predictive accuracy of those theories.
> In other words, we make progress on fine-tuning LLMs to provide better theoretical accounts of human behavior. We find that RL is particularly effective for shifting a prediction task into an explanation task by eliciting relevant CoTs from LLMs. We believe cognitive and neuroscientists could generalize this paradigm to other domains, with the aim of discovering useful theoretical insights through RL post-training that maximizes behavioral prediction.

---

> > ### Comment · Reviewer_Y1tw · 2025-11-27
> >
> > Thanks for the response.
> >
> > The authors mention that although this paper does not have novel algorithmic designs compared to machine learning community, it is a novel application and is a novel approach in cognitive science community. In that sense, I, as a machine learning person, cannot appreciate this paper from machine learning perspective. I will ask AC for advice.

---

### Official Review · Reviewer_2MpK · 2025-11-03

**Soundness:** 3
**Presentation:** 2
**Contribution:** 2
**Rating:** 4
**Confidence:** 4

**Summary:**

This paper proposes a reinforcement learning (RL)–based method to model human behavior in risky decision-making scenarios. The authors adopt the GRPO framework to investigate the potential of large language models (LLMs) to both predict decisions and explain the underlying decision process.

**Strengths:**

Strength:

1 The topic is interesting, and to my knowledge, this is the first work that applies RL to analyze the risky decision of human behavior.

2 The experiments are abundant.

3 Figure 1 provides a clear and concrete example that effectively illustrates the task.

**Weaknesses:**

Weakness:

1 Although the paper integrates RL to explain human decision-making and defines a reward function in Formula (1), it lacks an in-depth analysis of the task. As a result, the contribution appears incremental, and the work reads more like an experimental report than a research paper.


2  The paper leans more toward psychological or cognitive science research than computer science, as the main contributions involve cognitive interpretation rather than methodological innovation. Moreover, several cognitive science terms are used without sufficient explanation. For example, in the section “Cognitive mechanisms identified in CoT,” the term “cognitive mechanism” is not clearly defined.

**Questions:**

Suggestions and Questions:

1 I suggest that the authors provide clearer definitions and background information for the cognitive science terminology used in the paper, as noted in the weaknesses.

2 Regarding Figure 5: how do you determine the function of each representation pattern identified by t-SNE clustering? Specifically, how do you validate that a subcluster corresponds to concepts such as “consider risk preference” or “final estimates when EV difference is large”?

3 Could the authors provide a theoretical justification or formal analysis for the proposed reward decomposition?

---

> ### Author Response · Authors · 2025-11-25
>
> >Although the paper integrates RL to explain human decision-making and defines a reward function in Formula (1), it lacks an in-depth analysis of the task. As a result, the contribution appears incremental, and the work reads more like an experimental report than a research paper.
>
> **Response:** Thank you for your question! Could you please clarify the specification of ‘in-depth analysis of the task’ you would like to see? We are happy to add further details. We would also like to clarify our contributions are intended to be to the “Applications to Neuroscience & Cognitive Science” area of ICLR (https://iclr.cc/Conferences/2026/CallForPapers). In that spirit, the contributions are not intended to be new methods for RL training of LLMs, but rather a demonstration that applying those methods can result in meaningful innovations in how we can understand human cognition using AI.
>
> >The paper leans more toward psychological or cognitive science research than computer science, as the main contributions involve cognitive interpretation rather than methodological innovation. Moreover, several cognitive science terms are used without sufficient explanation. For example, in the section “Cognitive mechanisms identified in CoT,” the term “cognitive mechanism” is not clearly defined.
>
> >I suggest that the authors provide clearer definitions and background information for the cognitive science terminology used in the paper, as noted in the weaknesses.
>
> **Response:** In our context, the term “cognitive mechanism” refers to a verbalized theory that explains human risky choices – that is, the cognitive processes that drive people to reach a decision involving risk. We have now included the following text to clarify this concept: “In our context, cognitive mechanisms are specifically defined as verbal theories that explain how people reach a risky decision when presented with a particular set of options.”
>
> >Regarding Figure 5: how do you determine the function of each representation pattern identified by t-SNE clustering? Specifically, how do you validate that a subcluster corresponds to concepts such as “consider risk preference” or “final estimates when EV difference is large”?
>
> **Response:** For the category labels used in Figure 5, we examined the 10 thoughts closest to each centroid and provided an abstract summary based on those thoughts.
>
> >Could the authors provide a theoretical justification or formal analysis for the proposed reward decomposition?
>
> **Response:** Our reward function consists of two components: a correctness reward and a format reward. Maximizing the correctness reward encourages the model to predict the human data as accurately as possible, while maximizing the format reward ensures that the model follows the structured output format by placing the CoT before the behavioral prediction. The maximum format reward (0.5) is only half of the maximum correctness reward (1), indicating that the model is intended to prioritize correctness over format.
>
> The correctness reward is defined as 1 minus the absolute error between the model’s prediction and the human choice probabilities. This design choice is based on empirical observations that RL training is more stable (e.g., fewer KL spikes) under this reward compared to alternatives such as 1 minus the squared error or the negative cross-entropy loss. All of these correctness rewards have the desirable property that larger rewards correspond to better predictions, but the “1 minus absolute error” formulation proved to be more stable for our training setup.

---

### Official Review · Reviewer_nJgt · 2025-11-03

**Soundness:** 3
**Presentation:** 3
**Contribution:** 3
**Rating:** 6
**Confidence:** 4

**Summary:**

The authors propose a novel way in which LLMs can be used to aid cognitive science. They do RL fine-tuning on a Qwen model over a risky choices dataset and show that the fine-tuned model not only predicts choices better than cognitive models but also gives meaningful explanations for these choices.

**Strengths:**

The paper has a clear goal, and that goal is well executed. I think this type of modeling and analysis efforts will be interesting for the cognitive science community. The evaluations and ablations are quite comprehensive. The Appendix, in particular, contains several insightful analyses. The two that are very important for the paper’s message are:

- The ablation experiment in C.3, where the authors swap the CoT between the RL and the base models to show the importance of these traces.
- The experiment in D.3., where the authors show doing RL-finetuning on different datasets, that either contain random choices or choices from other data generative processes, change the CoT dramatically.

I also appreciate that the authors share some failed attempts in the appendix, which will be valuable to the research community.

**Weaknesses:**

I’m not convinced that RL is essential for this pipeline. The development of predictive and explanatory reasoning traces are attributed to RL by making comparisons to the base model. However, perhaps SFT (either Centaur style or full) is also sufficient to develop such traces. If this is the case, SFT may be preferred over RL fine-tuning given a) reduced computational costs during training and b) difficulties around getting RL to work, as the authors have also pointed out in section F.

**If the authors make the following comparisons between SFT and RL and show the benefits of RL under these setups, I will increase my score to 8. Otherwise, I will keep it at 6**:

- Generate reasoning traces from the SFT models. Plug these traces back into the base model. If RL is uniquely beneficial, the MSE of the SFT model here should be higher than 0.0212 (base model with RL reasoning traces).
- Replicate Figure 6a with the SFT model. Again, if RL is uniquely beneficial, i would expect the ordering of the cognitive mechanisms to look different for the SFT model.

My doubts originate from the findings of [this paper](https://arxiv.org/abs/2501.11120), where the authors show that doing SFT over risky choices gives models awareness about these choices. Similarly, in your setup, doing SFT can already give the models insights about the data generative process.

Minor:

- Experiments described in Appendix D.3. are very interesting. However, the results are only described qualitatively in a few sentences. Can you share more structured evidence here? e.g. something like the t-SNE plot or Figure 6a?
- At L 74, the text jumps too quickly into an example from the task. I’m familiar with the dataset, but for someone who is not, it would be helpful to have a sentence or two briefly describing what the task is.
- L 160 has a typo: a outcome-based

**Questions:**

- I appreciate the discussion of the elicitation hypothesis, suggesting the RL approach may ultimately be bound by the knowledge of the base model. If this is the case, the model can only interpolate within the existing knowledge. If so, how useful is the approach you are suggesting beyond being a proof concept? You point out that RL can help bring theories from other disciplines into psychology to generate knowledge. Cognitive scientists have been doing this for decades, therefore this approach still runs into the same limitations you discuss. I'd be curious to get your thoughts on this.
- In Figure 6a, the ordering and the proportions of different cognitive mechanisms seem to settle by the end of epoch 1. What else do you think is going on in the later parts of training that still allows the model to improve its predictions?
- How are the summary sentences in Figure 5 generated? Did you use GPT-4.1 for this too?

---

> ### Author Response · Authors · 2025-11-25
>
> >Generate reasoning traces from the SFT models. Plug these traces back into the base model. If RL is uniquely beneficial, the MSE of the SFT model here should be higher than 0.0212 (base model with RL reasoning traces).
>
> **Response:** This is a great suggestion, thank you! We implemented the following analysis: (1) we prompted the SFT and Centaur models to generate CoT before producing behavioral predictions, and (2) we removed the final predictions from the model completions and fed the remaining SFT- and Centaur-generated CoTs into the backbone LLM.
> We find that SFT-generated and Centaur-generated CoTs do not improve the backbone LLM’s behavioral predictions (see Appendix C.4 for details). This stands in contrast to the RL-generated CoTs, which do improve the backbone LLM’s predictions. We believe these results further strengthen the case for RL post-training to generate and refine CoTs that can subsequently enhance behavioral prediction.
>
> >Replicate Figure 6a with the SFT model. Again, if RL is uniquely beneficial, i would expect the ordering of the cognitive mechanisms to look different for the SFT model.
>
> **Response:** Thank you for the suggestion. We examined how the CoTs generated by the SFT and Centaur models change over the course of training epochs. Please see Figure 9 in the updated manuscript appendix for the visualization. Both SFT and Centaur-style finetuned models, when prompted to generate CoTs, show that the summarized cognitive mechanisms do not change substantially across training epochs. This further highlights the necessity of using RL to elicit more insightful CoTs.
>
> >Experiments described in Appendix D.3. are very interesting. However, the results are only described qualitatively in a few sentences. Can you share more structured evidence here? e.g. something like the t-SNE plot or Figure 6a?
>
> **Response:** Thank you. We have now included a visualization (similar to the original Figure 6a) of the evolution of cognitive mechanisms during RL training in the additional data control experiment, where complexity aversion is the data-generating behavior (see Figure 11 for details). We find that the relative rank of the minority thought (complexity aversion) steadily increases over the course of RL training.
>
> >At L 74, the text jumps too quickly into an example from the task. I’m familiar with the dataset, but for someone who is not, it would be helpful to have a sentence or two briefly describing what the task is.
>
> **Response:** Thank you. We have now revised the opening sentences to present the risky-choice problem more intuitively for readers.
>
> >L 160 has a typo: a outcome-based
>
> **Response:** Thanks, the typo was corrected!
>
> >I appreciate the discussion of the elicitation hypothesis, suggesting the RL approach may ultimately be bound by the knowledge of the base model. If this is the case, the model can only interpolate within the existing knowledge. If so, how useful is the approach you are suggesting beyond being a proof concept? You point out that RL can help bring theories from other disciplines into psychology to generate knowledge. Cognitive scientists have been doing this for decades, therefore this approach still runs into the same limitations you discuss. I'd be curious to get your thoughts on this.
>
> **Response:** Thank you for your thoughtful question. We agree that fine-tuning an LLM with RL may not introduce fundamentally new concepts, but rather helps surface, synthesize, and recombine existing knowledge. While it faces limitations similar to human scientist–driven discovery, RL post-training can substantially accelerate the pace of scientific exploration. This rapid and adaptive process allows us to update ideas, design new experiments, and propose new models and theories more frequently. In the long run, the CoTs and hypotheses generated through this process can inform future pretraining of base LLM models, which may further enhance the quality of elicited CoTs. For these reasons, we believe this approach could be beneficial to the community by accelerating the discovery of new scientific theories.

---

> > ### Author Response · Authors · 2025-11-25
> >
> > >In Figure 6a, the ordering and the proportions of different cognitive mechanisms seem to settle by the end of epoch 1. What else do you think is going on in the later parts of training that still allows the model to improve its predictions?
> >
> > **Response:** The frequencies of the top eight cognitive mechanisms identified in the CoTs do stabilize after one epoch of RL training. However, minor thoughts continue to be proposed and tested throughout training, although most of them never rise into the top eight. In other words, the RL process continues to explore the space of possible cognitive mechanisms, but the dominant mechanisms become stable after the first epoch.
> >
> > We also believe the model continues to learn how to more effectively leverage individual cognitive mechanisms over the course of RL training. For example, it learns to use “risk aversion” more appropriately by adjusting the degree of risk aversion in its reasoning to better capture human risky choices. This would indeed be a valuable direction for future research, particularly for mechanistically understanding how LLMs adjust their behavioral predictions based on CoTs.
> >
> > >How are the summary sentences in Figure 5 generated? Did you use GPT-4.1 for this too?
> >
> > **Response:** For the category labels used in Figure 5, we examined the 10 thoughts closest to each centroid and provided an abstract summary based on those thoughts. We did not use GPT-4.1 for this because these labels are more abstract than individual thoughts.

---

> > > ### Comment · Reviewer_nJgt · 2025-11-25
> > >
> > > Thanks for all the new analyses and the explanations. I'll update my score now

---

### Author Response · Authors · 2025-12-03

Dear AC,

To assist the final decision, we provide a summary of our work and the changes implemented during the rebuttal period.

Our work examines whether, and how, to best elicit chain-of-thoughts (CoTs) from LLMs that can both **explain** and **predict** human risky decisions. We show that outcome-based RL effectively adapts LLM-generated CoTs to match the structure of training data while maintaining strong predictive accuracy. Thus, RL post-training can augment conventional cognitive modeling approaches, where models typically aim only to maximize predictive accuracy; here, besides behavioral predictions, the CoTs elicited through RL become candidate verbal theories of human behavior.

Two initial reviews recommended acceptance (score 6), highlighting that “the paper has a clear goal, and that goal is well executed,” that “the evaluations and ablations are quite comprehensive,” and that “this type of modeling and analysis will be interesting for the cognitive science community” (nJgt). Another reviewer noted that our work “addresses an important and timely topic at the intersection of AI, behavioral economics, and cognitive sciences,” that “the use of CoT for explaining human choice is an elegant idea with potential applications that extend well beyond the scope of this paper,” and that “the experiments are comprehensive and rigorous” (ABiG).

The other two reviews were borderline (score 4) and negative (score 2), and comparatively brief. They expressed a desire for more innovations in the RL training algorithms themselves. While we did not introduce new RL algorithms, we conducted additional ablation and control experiments that clarify when and how RL post-training is effective for eliciting CoTs that explain human behavior, particularly in comparison to SFT and Centaur-style SFT.

A main suggestion was to control for the CoTs generated by SFT and Centaur-style SFT models. Although these models are not trained to produce CoTs, such CoTs can still be elicited after fine-tuning. To address this, we conducted new experiments in which we elicited CoTs from SFT models and then fed the CoTs (without the final JSON behavioral predictions) back into the backbone LLM. We found that these CoTs do not improve predictive accuracy (see the new Appendix C.4). This further supports the claim that RL training is needed to explore and refine CoTs in ways that meaningfully improve model predictions and enhance explanatory value.

We have also incorporated all other minor comments into the revision (highlighted in blue). We refer the AC to the detailed discussion below for further clarifications and context.

Kind regards,

The Authors

---

### Public Comment · ~Wenshuo_Wang5 · 2026-04-20
**The evidence seems stronger for prediction-supportive rationales than for faithful explanations**

Thank you for the interesting paper. I have one question about claim strength. As I understand it, the training signal directly rewards only the final predicted choice proportions (plus output format), rather than the correctness or faithfulness of the intermediate reasoning itself. The main quantitative evaluation also focuses on the final prediction target, while “explanation quality” is assessed mostly through downstream or post-hoc analyses such as human preference, LLM-as-judge, clustering, and CoT swapping.

My concern is that these results seem stronger as evidence that the method elicits rationales that are more useful for supporting accurate prediction than as evidence that the intermediate explanations are themselves more correct, more faithful, or genuinely better accounts of human decision mechanisms. In particular, CoT swapping appears to show primarily that the generated prefix is causally useful for downstream prediction, rather than that it constitutes a faithful explanation in a stronger sense.

If that reading is right, then the current evidence may fit a somewhat narrower conclusion more naturally: namely, that the method elicits more prediction-supportive, explanation-like rationales, rather than establishing higher-quality explanations in a stronger mechanistic or faithfulness-based sense. This leaves me wondering whether the present evidence is better understood as supporting a usefulness-oriented claim or a faithfulness-oriented one. Further clarification on this point could help readers better understand the intended scope of the paper’s explanatory claim.

---

### Meta-Review · Area_Chair_Ytiw · 2026-01-06

**Summary:**

This paper proposes a method to create human cognitive models that are interpretable and accurate by post training LLMs on human risky choices. Their chain of thought would verbalize the psychological concepts that lead to such choices. The results show that the RL-trained model achieved similar performance to the SFT one, but the reasoning patterns were significantly more interpretable. Interestingly, when replacing human data with synthetic data generated by an arbitrary reasoning rule, the RL-trained model was able to recover the correct reasoning patterns. The model is therefore not overfit to human cognitive patterns. The authors also show that a smaller model was not able to achieve the same performance due to procedural mistakes such as calculation errors that lead to incorrect conclusions.

Overall, most reviewers were positive about the paper. Reviewers 2MpK and Y1tw indicated that the paper was incremental from the perspective of contributions to ML algorithms, however precision was added that this paper is part of the "Applications to neuroscience & cognitive science" track. The concern is therefore rectified. An additional baseline was to compare the reasoning trace of RL-trained models to those of SFT-trained ones. The authors completed this and report that the chain of thoughts from SFT do not naturally evolve across training, and do not help when given as context to the base model. These results seem to support the claim that RL-generated CoT contain rich information that can improve behavioral prediction.

**Reviewer Concerns:**

One of the concern shared by Reviewers 2MpK and Y1tw was that the paper proposed only incremental contributions in terms of ML algorithms. This was resolved through a conversation with the AC who rectified that this paper was part of the "Applications to Neuroscience & Cognitive Science" track.

Reviewer nJgt further had concerns over the necessity of using RL as opposed to a simpler baseline like SFT. The reviewer stated in their original review that they would raise their score if experiments with SFT were not successful. The authors conducted these experiments and reported that indeed RL was necessary.

Finally, Reviewer ABiG argued that a simple baseline from ML explainability methods would be important here. The authors noted that this relied on selecting the right features. It is unclear if the reviewer would have been content with the rebuttal and would have adjusted their score.

**Reviewer Scores:**

Reviewer nJgt: would have raised their score.
Reviewer 2MpK: unlikely to raise score.
Reviewer Y1tw: very unlikely to raise score.
Reviewer ABiG: it is possible they would have raised their score.

---

### Decision · Program_Chairs · 2026-01-26

Accept (Poster)